# The Evolutionary History of New Zealand *Deschampsia* Is Marked by Long-Distance Dispersal, Endemism, and Hybridization

**DOI:** 10.3390/biology10101001

**Published:** 2021-10-05

**Authors:** Zhiqing Xue, Josef Greimler, Ovidiu Paun, Kerry A. Ford, Michael H. J. Barfuss, Jorge O. Chiapella

**Affiliations:** 1Department of Botany and Biodiversity Research, Faculty of Life Sciences, University of Vienna, Rennweg 14, 1030 Vienna, Austria; ovidiu.paun@univie.ac.at (O.P.); michael.h.j.barfuss@univie.ac.at (M.H.J.B.); 2Allan Herbarium, Manaaki-Whenua Landcare Research, P.O. Box 69040, Lincoln 7640, New Zealand; FordK@landcareresearch.co.nz; 3Instituto de Investigaciones en Biodiversidad y Medioambiente (INIBIOMA-CONICET-Universidad Nacional del Comahue), Quintral 1250, Bariloche R8400FRF, Argentina; jchiapella@comahue-conicet.gob.ar

**Keywords:** *Deschampsia*, New Zealand, endemics, hybridization, RADseq, plastid sequences, morphology

## Abstract

**Simple Summary:**

The evolution of the flora of New Zealand has been the subject of many debates, including trans-oceanic dispersal and vicariance. *Deschampsia*, which is a grass genus distributed in cold–temperate regions of both hemispheres, includes the cosmopolitan *D. cespitosa* and several endemic species in New Zealand. We applied Restriction Site Associated DNA sequencing, plastid genome, and morphological data to study the evolution of *Deschampsia* in New Zealand. Native New Zealand *D. cespitosa* was found in a northern hemisphere clade together with samples of Europe, Canada, Russia, China, Korea, whereas the three endemics were found in a southern hemisphere clade, together with *D. antarctica* of Argentina and Antarctica. We hypothesize that the endemics diverged from a common ancestor with *D. cespitosa* in the late Miocene or Pliocene. Hybridization between *D. cespitosa* and the endemic *D. chapmanii* was confirmed. Our study qualifies as a model of thorough analyses for a non-model group and provides evidence for the evolutionary history of this widespread genus in New Zealand.

**Abstract:**

The contrasting evolutionary histories of endemic versus related cosmopolitan species provide avenues to understand the spatial drivers and limitations of biodiversity. Here, we investigated the evolutionary history of three New Zealand endemic *Deschampsia* species, and how they are related to cosmopolitan *D. cespitosa*. We used RADseq to test species delimitations, infer a dated species tree, and investigate gene flow patterns between the New Zealand endemics and the *D. cespitosa* populations of New Zealand, Australia and Korea. Whole plastid DNA analysis was performed on a larger worldwide sampling. Morphometrics of selected characters were applied to New Zealand sampling. Our RADseq review of over 55 Mbp showed the endemics as genetically well-defined from each other. Their last common ancestor with *D. cespitosa* lived during the last ten MY. The New Zealand *D. cespitosa* appears in a clade with Australian and Korean samples. Whole plastid DNA analysis revealed the endemics as members of a southern hemisphere clade, excluding the extant *D. cespitosa* of New Zealand. Both data provided strong evidence for hybridization between *D. cespitosa* and *D. chapmanii*. Our findings provide evidence for at least two migration events of the genus *Deschampsia* to New Zealand and hybridization between *D. cespitosa* and endemic taxa.

## 1. Introduction

The New Zealand archipelago in the southwestern Pacific, centered on three main islands, with smaller adjacent islands, is isolated by more than 1500 km from other large land masses. The origin and evolution of the flora of New Zealand has been the subject of debates regarding the relative contributions of trans-oceanic dispersal [1,2,3,4,5,6] versus vicariance [7,8,9,10,11]. There is strong evidence from the fossil record [3,5,12] and molecular dating [13,14,15,16,17,18,19] that trans-oceanic dispersal is an important and omnipresent feature that played a key role in shaping the New Zealand flora [15,20,21] resulting in species radiations [14,22,23,24,25], but also clades with limited speciation [26,27].

It is also likely that trans-oceanic dispersal and subsequent diversification has shaped the New Zealand flora [4,6,20,21]. These studies argue that there is little evidence for direct Gondwanan ancestry in the present flora of the archipelago, and even the Gondwana icon Nothofagaceae is likely to have colonized New Zealand more recently [20,21,28]. Given that New Zealand is a moderately sized archipelago that is a remnant of the large, submerged continent of Zealandia, it is not surprising that numerous biogeographic patterns have been observed exacerbating generalizations [20,21]. Both climate oscillations and geological events such as rapid uplifting of mountain ranges, especially during Pliocene and Pleistocene, have had a strong impact on the extant New Zealand biota.

Australia, South America and Eurasia have been previously proposed as sources of migrants to New Zealand [6,15,29,30,31]. Dispersal from the Northern to the Southern hemisphere across the tropics has most likely occurred in some cool-season grass species (*Agrostis, Festuca, Koeleria* and *Trisetum*, and sedges [30,32].

Migration from Eurasia to Australasia may have also included the mountains of New Guinea, where, e.g., the endemic *Deschampsia klossi* thrives in the subalpine belt (3300–4000 m) of Mt. Wilhem [29,30,33]. Dispersal to Australia and New Zealand bypassing the tropical archipelagos was supposed by Wardle (1978) [30] for *D. cespitosa* and two alpine species of *Carex* (*C. lachenalii* and *C. pyrenaica*).

Here, we focus on the grass genus *Deschampsia* P. Beauv in an endemic-rich region of the southern hemisphere. The genus, belonging to the cold–temperate core Pooideae, most probably evolved and radiated in landmasses of the Northern Hemisphere, i.e., Eurasia or North America, the likely ancestral area of core-Pooideae [34], but it is also found in cold and cold–temperate regions of both hemispheres [35,36]. Extant populations of the cosmopolitan *D. cespitosa* in the Southern Hemisphere are considered ancient migrants [37,38] and thus regarded as native. To our knowledge there is no investigation involving a global sample of *D. cespitosa* or a phylogeny of the genus involving a worldwide sample, although there are several investigations covering larger regions using morphological [39,40] or molecular approaches [41,42].

Several endemic *Deschampsia* species have evolved worldwide, especially on islands, e.g., on Tristan da Cunha (*D. christophersenii, D. mejlandii, D. robusta, D. waceii*), Hawaii (*D. nubigena*), Madeira (*D. argentea, D. maderensis*), the Azores (*D. foliosa*), and Papua New Guinea (*D. klossi*). Other endemics are found in high mountains, such as the Pamir (*D. koelerioides, D. pamirica*), Central Mexico (*D. liebmanniana*), and eastern tropical Africa (*D. angusta*). These mountains function as ‘islands’, since species found atop them are isolated from the nearest relatives by lowlands acting as barriers to dispersal [43,44]. 

In New Zealand, the genus is represented by five species [45]: the cosmopolitan *Deschampsia cespitosa* and four endemic taxa, *D. chapmanii* Petrie, *D. gracillima* Kirk, *D. pusilla* Petrie, and *D. tenella* Petrie. Hybridization between the endemics *D. chapmanii* and *D. tenella* [46] has been assumed based on herbarium vouchers, as well as between *D. cespitosa* and the endemics by field observation (K.F, pers. obs.).

*Deschampsia cespitosa* is widespread in New Zealand and is found on the three main islands, North, South and Stewart Island, and on two offshore islands, Auckland and Chatham. It is a species of lowland to subalpine wetland habitats and is primarily found in marshes near the coast in Southland, Westland and the offshore islands. It is also commonly found scattered through the intermontane basin river/lake systems of the eastern South Island, and on the edges of tarns and lakes in the Southern Alps up to 1200 m a.s.l. It is less common in the north of the South Island and the North Island, and it is considered in decline and threatened [47] as in earlier times it was recorded in abundance from the lower Waikato southward [48,49,50]. *Deschampsia cespitosa* in New Zealand does not form large widespread swards and it is not a dominant component of montane and alpine tussock-grasslands as are species of other grass genera such as *Chionochloa*, *Festuca* or *Poa* [45,51]. However, Mark and Dickinson (2001) [38] proposed a new indigenous vegetation type of ‘*Deschampsia cespitosa* subalpine tussockland’ based on a localised sward on a terrace at a stream entrance to the colloquially named ‘Pyramid’ lake in Fiordland.

All four endemic *Deschampsia* species of New Zealand are found in damp sites in forests, or in shrubland from near sea level, and up to higher altitudes in tussock-grassland and herbfield [45]. They are all smaller tufted species compared to the prevailing tussock habit of *D. cespitosa* and are found in associations with other species. *D. chapmanii* and *D. tenella* are common and widespread—the former is found in herbfields, kettlehole bogs and alpine turfs and the latter in montane to subalpine forest and shrubland in gaps and along margins. *D. gracillima* is restricted to the Subantarctic Islands, including the Auckland and Campbell Islands, and to southern Fiordland, whereas *D. pusilla* is a minute high alpine species of herbfield localised to mountain ranges in Central Otago.

Here, we used Restriction Site Associated DNA sequencing (RADseq) and whole plastid sequences, together with morphological data to investigate the evolutionary history of the genus in New Zealand. In detail we investigate: (i) The phylogenetic relationships among the cosmopolitan *Deschampsia cespitosa* and three endemic taxa (*D.*
*chapmanii, D. gracillima* and *D. tenella*); (ii) the nature and origin of putative hybrids between *D. cespitosa* and endemics in New Zealand; (iii) the relationship of *D. cespitosa* of New Zealand to *D. cespitosa* from outside the archipelago (including samples from Australia and other regions).

## 2. Materials and Methods

### 2.1. The Study Species

All *Deschampsia* species investigated here are perennial plants with small spikelets and (mostly) two florets. Based on genome size estimates [52] of representative samples (except for those where plastid sequences were retrieved from Genbank), the populations involved in this study are deemed diploid (2*n* = 2*x* = 26). 

### 2.2. Taxon Sampling

In this research we included 57 individuals for the RADseq analysis: eleven *Deschampsia* populations, i.e., sampling localities) from New Zealand, one population from Korea (five samples) and one population of Australia (herbarium material, two samples from Macquaire Island). According to the morphological identification, our samplings from New Zealand included: *Deschampsia tenella* (one population from the ‘Pyramid’ lake site)*, D. gracillima* (two populations)*, D. chapmanii* (two populations), and *D. cespitosa* (five populations; seven individuals from the ‘Pyramid’ lake site). Several attempts to collect *D. pusilla* failed. One sampling locality appears with three population numbers because based on morphological characters it was considered a priori to consist of *D. cespitosa* (population 72), a hybrid between the former and *D. tenella* (71) and another swarm of possible hybrids (73). This ‘Pyramid’ lake terrace site (167.366992 E–45.746661 S) is shown in Appendix A. Vouchers of each species are deposited in the herbarium of the University of Vienna (WU) and the Allan Herbarium (CHR) of Manaaki-Whenua, Lincoln. The map of sampling locations in New Zealand and Australia was generated with ArcGis (Figure 1A). 

We included 26 individuals on a worldwide scale for the whole plastid sequencing. A sample of each New Zealand population was selected. In addition, one to five samples each of *D. cespitosa* of Australia (New South Wales), Argentina (both *D. antarctica* and *D. cespitosa*), Austria, Estonia, Iceland, the United Kingdom, Canada, Russia (including both European and Asian individuals), Korea, and China (*D. cespitosa* subsp. *pamirica* and *orientalis*, *D. koelerioides*) were analysed. Detailed information on all samples is given in Appendix A. The map (generated with ArcGis) of sampling locations for the whole plastid sequencing is given in Figure 1B.

### 2.3. DNA Extraction, Library Preparation, and Illumina Sequencing

Total genomic DNA was extracted from silica-dried leaves (20 mg) by Invisorb Spin Plant Mini Kit (STRATEC Molecular GmbH, Berlin, Germany). DNA was purified using the Nucleospin gDNA clean-up Kit (MACHEREY-NAGEL GmbH, Düren, Germany). The DNA content was then quantified using a Qubit 3.0 Fluorometer and the dsDNA HS Assay Kit (Thermofisher Scientific, LifeTech Austria Zweigniederlassung, Vienna, Austria). 

Single-digest RADseq libraries were prepared for 48 individuals in each library, following the protocol given in Paun et al. (2016) [53] with some modifications as follows: 200ng DNA and 15 U PstI restriction enzyme was used per sample, and later 300 nM P1 adapters were ligated to the digested samples. The DNA was sheared with a Bioruptor Pico and 2 cycles of 45 sec ON and 60 sec OFF at 4 °C. Six base-pairs inline and index barcodes with at least three different positions were used. All libraries were sequenced as paired-end 125 bp reads on an Illumina HiSeq 2500 at the Next Generation Sequencing Facility at Vienna BioCenter Core Facilities (VBCF), Austria.

An additional NGS library was prepared for sequencing the plastid genomes from 20 *Deschampsia cespitosa* samples, one hybrid, one *D. koelerioides*, one *D. antarctica* and one individual of each of the endemics. The preparation was performed using the TruSeq DNA PCR-Free Low Throughput Library Prep Kit (Illumina Inc, San Diego, CA, USA). We used a Biorupter Pico sonication device to shear 1000 ng DNA from each sample with eight cycles of 15s on and 90s off at 4 °C. An average targeted fragment size of 350 bp for each library was prepared following the manufacturer’s protocol. All individual libraries were then pooled with equal representation to a final library. The library was sequenced as 50 bp paired-end reads on an Illumina HiSeq 2500 at NGS Facility of VBCF.

### 2.4. SNP Filtering

The raw bam files were first evaluated with FastQC v.0.11.9 (available from https://www.bioinformatics.babraham.ac.uk/projects/fastqc/, accessed on 18 May 2020). After checking reads number and quality, the raw bam files were demultiplexed to sublibraries first based on index reads using BamIndexDecoder v.1.03, which is included in the Picard Illumina2Bam package (available from https://github.com/wtsi-npg/illumina2bam, accessed on 1 April 2019). Each sublibrary bam file was further converted to paired-end fastq files with SamToFastq tool in Picard tools v.2.18.26 (available from https://github.com/broadinstitute/picard/, accessed on 1 April 2019). These fastq files were further processed with Stacks v.1.47 [54], starting with process_radtags, which is used to demultiplex individuals based on inline barcodes. Simultaneously, quality filtering was performed to remove any reads with uncalled bases and those with low average quality scores by rescuing barcodes and restriction sites with a maximum of one mismatch.

Each demultiplexed file was further mapped to a published reference of *Hordeum vulgare* (GenBank assembly accession: GCA_004114815.1 [55]) using BWA v.0.7.12-r1039 [56] with default settings, after first indexing the reference using the option ‘-a bwtsw’ for a long reference genome. The aligned sam files were sorted by reference coordinates, and read groups were added into their headers with Picard tools. The outputted bam files were further indexed with SAMtools v.1.6 [57] and realigned around indels with the Genome Analysis Toolkit v.3.81 [58]. Genotypes were then called with ref_map.pl in Stacks using default settings, including a minimum number of three identical reads to retain a stack. The program populations from Stacks, was further used to produce vcf and haplotype files, retaining only loci with an observed heterozygosity of 0.65 or less, to avoid using further any pooled paralogs. The program vcftools v.0.1.5 [59] was used to filter the resulting dataset with various settings. We retained SNPs present in at least 75% and 90%, respectively, of individuals. Missingness and inbreeding coefficient for individuals were calculated with vcftools and visualized for each species as vioplots in R with Rstudio v.1.1.463 [60]. The filtered vcf file was further converted to other formats (phylip and nexus) with PGDSpider v.2.0.8.2 [61].

Besides, ANGSD v.0.929-24 [62] was also used to calculate genotype posterior probabilities based on the realigned mapped RADseq bam files. We adjusted -minInd to 50% individuals, -minMaf for frequency of two individuals, mapping quality threshold of 20, base quality threshold of 20 and retained only variable positions with a high confidence (*p*-value < 1 × 10^−6^). 

For reconstructing whole plastid sequences, NGS whole-genome data was demultiplexed based on index reads, allowing for a maximum of one mismatch with the program BamIndexDecoder. Then, the paired-end raw reads of each individual were de novo assembled with the program FAST-PLAST v.1.2.6 (https://github.com/mrmckain/Fast-Plast, accessed on 18 May 2020). Trimmomatic v.0.36 [63] was used to filter the raw reads: we used the option ILLUMINACLIP to remove adapters. The rest of the settings followed Heckenhauer et al. (2019) [64]. Quality-filtered reads were then mapped back to the assembled plastome with BWA to check if the coverage was even along the length of the respective reference. Based on the mapping results, the sequences were manually corrected in IGV v.2.3.68 [65], especially around the junctions between the single copy regions and the inverted repeats. This pipeline was able to reconstruct the complete plastid genomes only for two of the 26 individuals (accession 337 and 345 from New Zealand). For all individuals, we used a mapping-based pipeline [64] combining BWA, and GATK HaplotypeCaller for haplotypes to extract the plastid genome SNP information as a vcf file, then replaced the reference with the SNP information with GATK FastaAlternateReferenceMaker. For this analysis, we used *Deschampsia cespitosa* (NC_040999) as the plastid reference. 

### 2.5. RADseq Evolutionary Analyses

To firstly assess the relationships between different populations and individuals, we used ANGSD genotype posterior probabilities to calculate a covariance matrix in PCAngsd v.0.981 [66] and then plotted it as a PCA using the R package Ggplot2 v.3.2.1 (https://ggplot2.tidyverse.org, accessed on 18 October 2019) in Rstudio. The covariance matrix was also plotted as a heatmap with the R package Gplots v.3.1.1 (http://CRAN.R-project.org/package=gplots, accessed on 30 March 2021) in Rstudio. Admixture was estimated with NGSAdmix v.33 [67], running the program from K = 1–9, with ten independent runs for each K. Finally, Evanno’s ΔK method [68] was used to choose the best likelihood in Clumpak [69]. A phylogenetic network based on a nex file including maximum 25% missing data, was estimated in SplitsTree v.4.14.8 [70]. Based on the concatenated polymorphic loci, we also constructed phylogenetic trees (i.e., with the hybrid population included and excluded, respectively) with RAxML v.8 [71] in CIPRES website [72]. The settings were as follows: 1000 rapid bootstrap replicates when searching for the best-score maximum likelihood (ML) tree, the GTRCAT model and an ascertainment bias correction of the likelihood by Lewis (2001) [73] as recommended for concatenated SNP datasets. We used the information from *Hordeum vulgare* extracted from the vcf file as outgroup. Tree results were visualized with Figtree v.1.4.4 (https://github.com/rambaut/figtree/, accessed on 23 October 2019) and were rooted using *Hordeum vulgare* as outgroup.

Prior to constructing a species tree, we performed Bayesian Species Delimitation analyses (BFD). In order to decrease computational costs BFD were performed on a reduced dataset including only 15 samples, i.e., at least three individuals for each *Deschampsia tenella*, *D. gracillima*, *D. chapmanii* and *D. cespitosa*. The vcf files which allowed for maximum 10% missing data and only included single SNPs for each RAD tag were converted to phylip and nexus format with PGDSpider. The nex file was used to create the BFD input xml files in BEAUti v.2.4.5 [74]. Five different models of species assignments were tested, combining differently or leaving as distinct units each of the endemics. The BFD analyses were performed in SNAPP v.1.2.5 [75] using 12 initialization steps and one million chain-lengths for each clades model. The rest of the settings followed Brandrud et al. (2020) [76]. 

Based on the BFD results, the same dataset was used to further build a species tree with SNAPP. Chain-length was set to ten million, saving a tree every 1000 generations. The rest of the settings again followed Brandrud et al. (2020) [76]. The log files were evaluated for convergence of the SNAPP analysis with Tracer v.1.6 [77]. The SNAPP trees were viewed as a cloudogram with Densitree v.2.2.6 [78] after removing the first 10% trees as burn in. In order to infer minimum and maximum divergence times between different species, the SNAPP tree was calibrated with a general mutation rate of 6.03 × 10^−9^ substitutions per site per generation estimated for Poaceae [79,80], and assuming a generation time of between two and five years, previously estimated for the closely related *Deschampsia antartica* [42]. The results were rescaled in Densitree based on the total number of investigated sites and the number of polymorphic sites across this investigated length.

To assess the phylogenetic relationships and admixture among all species (including also the putative hybrid population), a Treemix v.1.13 analysis [81] was performed using the unlinked dataset (i.e., one SNP per locus). The vcf file was obtained by vcftools, and converted to the Treemix format using a Python script (https://github.com/wk8910/bio_tools/tree/master/03.treemix, accessed on 7 May 2020). The analysis excluded the Australian accessions of *D. cespitosa* due to the high proportion of missing data (Appendix A). Treemix was run incrementally allowing for migration events and we then compared the increase in the total variation explained to infer how many migrations to allow. The tree was visualized in R. 

### 2.6. Phylogenetic Analyses for Whole Plastid Genomes

We choose *Holcus lanatus* (GenBank accession number KY432781) and *Dactylis glomerata* (NC_027473) as outgroups. We also included published sequences of *Deschampsia antarctica* (NC_023533) in the whole plastid genome alignment. The nexus files were further used to build a maximum parsimony tree in PAUP v.4.0a [82]. We used heuristic search via stepwise addition, the number of replicates set to 1000 and addition sequence set to random. We then conducted bootstrap analysis (1000 replicates). In addition, a ML tree was constructed in RAxML using 1000 rapid bootstrap replicates and the GTRGAMMA model. Both trees with bootstrap values were finally visualized with FigTree, using *Holcus* and *Dactylis* for rooting.

### 2.7. Morphology

For eight to eleven individuals per taxon we measured basic morphological features (plant height, panicle length and width, penultimate and basal leaf length, lower and upper glume length, lemma length, awn length; see Appendix A), previously used as key characters to differentiate species of *Deschampsia* [45]. Two non-parametric analyses were carried out, a Kruskall–Wallis including all taxa and a Mann–Whitney U-test for pairs of *Deschampsia* taxa (*cespitosa*/*chapmanii, cespitosa*/*gracillima, cespitosa*/*tenella, chapmanii*/*gracillima, chapmanii*/*tenella,* and *gracillima*/*tenella*). We applied Bonferroni corrections to all significance levels recovered, to prevent the problem of Type I errors in multiple comparisons. Box plots of selected characters were finally drawn. All tests and figures for morphological data were performed in R.

## 3. Results

### 3.1. RADseq

After removing uncalled and low-quality reads, the average number of raw RAD reads per individual was 4.4 million. The individual read files had an average mapping success to the reference genome of *Hordeum vulgare* of 33% and an average coverage of 8.2× (i.e., ranging from 4.7× to 13.8×). After allowing a maximum observed heterozygosity 0.65 in the Populations program and 25% missing data in vcftools, we retained 17,342 SNPs, whereas the dataset including only unlinked SNPs had 6127 positions. Allowing maximum 10% missing data, the unlinked SNPs dataset included 1600 positions. The inbreeding coefficient per individual (F) was positive as a signal of inbreeding for the Korean and Australian *Deschampsia cespitosa* groups. However, F was found to be mildly negative in *D. gracilis* and *D. chapmanii*, and took the maximum negative value in the putative hybrid population (Appendix A). *Deschampsia tenella* and New Zealand *D. cespitosa* showed on average inbreeding coefficients close to zero.

After filtering, ANGSD retained 621,138 confidently variable sites (total number of sites reviewed: 55,479,806). The genotype-free ANGSD-derived PCA result including 57 individuals presented a clear distinction on the first two axes, explaining 11.9% of the total variation (Appendix A). The individuals were separated into seven lineages: the first included all five populations of cosmopolitan *D. cespitosa*, the three endemic species (*D. gracillima, D. tenella, D. chapmanii*) each separated into a different lineage, a further cluster included the assumed hybrid population (based on morphological characters), and the last two clusters were represented by the Korean samples of *D. cespitosa,* and, respectively, by the two Australian accessions. The relatedness heatmap (Figure 2A) provided a detailed view of these relationships and showed that the hybrid population rather had highest coancestry to *D. cespitosa* and *D. chapmanii*. Moreover, the heatmap also showed a closer link of the endemics *D. chapmanii* and *D. tenella* to the Australian and, respectively, Korean populations of *D. cespitosa* than to those of New Zealand. 

The patterns observed in the PCA and coancestry heatmap were further confirmed by the NGSadmix results. Evanno’s ΔK statistic supported K = 2 as the best K value (Appendix A), which clearly separated *Deschampsia cespitosa* on one side, and the three New Zealand endemic species. The link between the endemics and *D. cespitosa* of Australia and Korea was also indicated by this model, whereas the origin of the hybrid population was confirmed by all models. Besides, K = 3, 5, 6 showing notable ΔK peaks were also informative (Figure 2B). Increasing K identified further structure within species and regions; however, none of them separated the Australian and Korean samples. 

The SplitsTree phylogenetic networks based on RADseq revealed significant reticulation pattern close to the origin of the three endemic species (*Deschampsia gracillima, D. tenella, D. chapmanii*) (Appendix A). The hybrid population was placed in the middle of the network, in between the endemic species and the cosmopolitan species.

The midpoint rooted RAxML tree (Appendix A) shows two highly supported major clades (bootstrap percentage—BP 100). The first major clade contains all samples of *Deschampsia cespitosa* with the Australian and Korean samples forming a clade that is sister to a New Zealand clade. The other major clade includes the endemic species, with each of them forming highly supported (BP 100) clades. After adding the hybrid population, which connected to the *D. cespitosa* clade with low support (BP 68), the basic bootstrap support within *D. cespitosa* in general decreased, whereas the other relationships remained unchanged (Appendix A).

The highest marginal likelihood estimate, and hence the best species delimitation model distinguished the four species (*Deschampsia cespitosa, D. chapmanii, D. tenella, D. gracillima*) as independent units (Figure 3A). For the SNAPP species tree of these four species, the topology was similar with the RAxML tree, with *D. chapmanii* being shown to be sister to a clade composed by the two remaining endemics (Figure 3B). The output log file was evaluated in Tracer, showing that all ESS values for this analysis were above 200. According to the range of generation times (i.e., between two and five years), the rescaled SNAPP tree indicated that the divergence time between *D. cespitosa* and the three endemic species lies between 3.8 Ma and 9.5 Ma, whereas the split of *D. chapmanii* from the ancestor of the other two species happened about 2.6 to 6.5 Ma, and *D. tenella* and *D. gracillima* diverged about 1.8 to 4.8 Ma. 

The general topology of the Treemix ML tree (Figure 3C) was consistent with the phylogenetic relationship recovered from the other phylogenetic analysis. One migration event between the hybrid population and *Deschampsia chapmanii* improved the explained variation from the initial 93.3% (with zero migration events allowed) to 99.5% (with one migration event allowed). Further migration events (two to four) improved the variation by ′0.01%, 0.4% and 0.01%. As a result, the scenario with only one migration event was chosen to be the best model. 

### 3.2. Whole Plastid Data

The number of raw reads in the NGS whole genome sequencing after demultiplexing ranged from 2.3 to 15 million, with an average of 7.8 million. However, only 17,480–194,221 pairs (average 91,759) could be mapped to the plastid reference. Among the 26 individuals, only two accessions assembled the complete plastid sequences as a contiguous sequence with the FAST-PLAST program (accession 337 and 345 from New Zealand). After mapping back and checking the coverages along the genome, finally, the lengths of these two genomes were 135,318 bp and 135,326 bp, respectively. The GC content of both accessions was 38.3%. Mapping the filtered reads of all individuals to the published *Deschampsia cespitosa* (NC_040999) resulted in an average coverage from 7× to 73× (average 35×). The Australian herbarium sample had the lowest average coverage (7×). 215 polymorphic positions were recovered between the 26 *Deschampsia* accessions. After adding the outgroups, the final matrix included 137,498 characters, while the potentially parsimony informative sites were 3557 characters. 

The topologies of ML (Figure 4) and MP (not shown, bootstraps are provided on the branch of ML tree) trees were congruent. Both trees confirmed the clear separation between the endemics and *D. cespitosa* in New Zealand. The hybrid accession was nested within New Zealand *D. cespitosa.* Both trees split into two main clades: the southern clade and northern clade. The southern clade includes the three New Zealand endemics and *D. antarctica* of Argentina and Antarctica. The other clade includes *D. cespitosa* of New Zealand and (incl. *D. koelerioides*) all other regions. In detail, we found five genetic groups corresponding to Europe (1); Argentina (2); Qinghai–Tibet Plateau of China (subsp. *orientalis*) (3); New Zealand and Australia (4); China (Xinjang), Korea, Russia (Asia) and Canada (5). The European individuals were firstly separated from this second major clade, followed by samples of Argentina, the Qinghai–Tibet Plateau and a poorly supported rest containing an Australia–New Zealand *D. cespitosa*-clade and an Asian clade (without Qinghai–Tibet). 

### 3.3. Morphology

The differences in morphological variation of the selected characters can be seen in the boxplots (Figure 5 and Appendix A). A Kruskall–Wallis test (Appendix A) showed highly significant differences (*p* < 0.001) for all characteristics analysed except for panicle width and awn length (*p* < 0.05). The Mann–Whitney pairwise comparisons depicted in Table 1 showed that nearly all vegetative characters (plant height, basal and penultimate leaf length) differed significantly between *Deschampsia cespitosa* and each of the endemic taxa, except for the penultimate leaf length, which could not distinguish between *D. cespitosa* and *D. tenella*. The reproductive characters (panicle and spikelet) also differed in most pairwise comparisons; specifically, glume length differed in all comparisons except for *D. chapmanii/D. tenella* and awn length differed in all cases except for the comparisons between the endemics. In the voucher of population 73 confirmed by genomic data as hybrid between *D. cespitosa* and *D. chapmanii* we found characters of the spikelet (length of glumes, lemma, and awn) intermediate between *D. cespitosa* and *D. chapmanii*, while the awn insertion pointed to the latter.

## 4. Discussion

### 4.1. Relationships between Deschampsia cespitosa and the Endemics

We found high support from both RADseq and plastid data for the separation of the three New Zealand endemic taxa (*Deschampsia tenella*, *D. gracillima*, *D. chapmanii*) as distinct species. The RADseq and the plastid data point to an evolutionary scenario where the endemics in New Zealand have diverged from an ancestral southern lineage. They appear not to be closely related to the *D. cespitosa* existing at present in New Zealand, whereas they obviously share a portion of the ecological niche with this taxon [45] particularly in more southern latitudes of the South Island and Stewart Island. Our morphological data also suggest an overall similarity of the endemic taxa as opposed to the cosmopolitan *D. cespitosa,* especially in reproductive characters, which is in line with Edgar and Connor (2010) [45]. We found the morphological differences between the endemics most pronounced in comparing *D. gracillima* and *D. tenella*.

Our divergence time estimates from the coalescent analyses (Figure 3B) between *Deschampsia cespitosa* and the endemics point to a splitting off from a common ancestor in the late Miocene or Pliocene followed by subsequent cladogenesis in the lineage leading to the endemic taxa, the latest with the onset of the Pleistocene. The Pleistocene also marks the appearance of many other plant groups of open habitats, some species-rich [14,22,23,24,25] and correlates with the formation of tall mountains and alpine conditions in the Southern Alps [21,31,83]. The resulting diverse alpine and montane habitats are also the likely trigger for speciation in the endemic *Deschampsia* lineage. There is no solid evidence on when and how an ancient “*D. cespitosa*” arrived in the region Australis. Wardle (1978) [30] argues that the present *D. cespitosa* as well as two sedges made its way from the northern hemisphere to this region, bypassing Malaysia and New Guinea. The many confirmed locations around the globe with endemic *Deschampsia* taxa [36], most likely derived from an ancient “*D. cespitosa*”, suggest an evolutionary history which included several dispersal events across oceans and huge landmasses, resulting in the establishment of many populations of the cosmopolitan species in the temperate zones of both hemispheres. Depending on the biological features of the taxon, time, and the ecological setting in the new environment, this can result in anagenetic or cladogenetic divergence [84] and emergence of endemic taxa, as is the case in New Zealand.

Present *Deschampsia cespitosa* and ancient “*D. cespitosa*”, as other grasses of the Eurasian BEP clade [34], are among the most successful organisms on our planet. The grass is obviously able to migrate over long distances due to several biological features. A single well grown tussock with many panicles may produce up to half a million dispersal units (karyopses) which can effectively be dispersed by wind and may remain viable for several years [85]. Such prerequisites may result in several long-distance dispersals giving rise to distant divergent populations leading to endemic lineages, which add a high portion to global biodiversity.

### 4.2. The Hybrids at the ‘Pyramid’ Lake Terrace Site

The maternal lineage for the hybrid between *Deschampsia cespitosa* and the endemic *D. chapmanii* is most likely the former, assuming that plastid transfer via pollen is a rare event in grasses or appears only at very low frequencies [86]. The parents were a priori thought to be correctly *D. cespitosa* and, erroneously, *D. tenella* (as the rachilla in the voucher specimen thought to be the other parent is hairy, whereas the rhachilla in *D. chapmanii* is glabrous). *Deschampsia chapmanii* in our study was not collected amongst or even near the sampled hybrid population. At this site hybrid plants appear to dominate the terrace, with larger *D. cespitosa* plants growing amongst the lower statured hybrid sward (Appendix A). However, the presence of *D. chapmanii* in the greater area is confirmed by specimens in New Zealand herbaria (CHR, OTA). The constitution of the hybrid may point to high pollen pressure from the endemic *D. chapmanii*. From this one can assume that *D. cespitosa* is, in general, less abundant than the endemic. In fact, at this locality and towards its altitudinal limit in the south (740 m a.s.l.) it is especially scattered in distribution [38,45] (AK, CHR Herbarium records). This is also where it is likely to come into contact with either *D. chapmanii* or *D. tenella*, more likely *D. chapmanii* as they are both plants of open areas. From our data (nearly balanced proportions in Figure 2B) we conclude that this is a rather recent hybridization because for an ancient introgression we would expect a shift towards the dominant taxon. Although no evidence was detected of hybridization in other populations sampled, there is some indication from New Zealand herbarium specimens that there may be other sites where two species meet and hybridize (CHR 402550, 394444).

### 4.3. Relationships of New Zealand Deschampsia Species to Those of Other Regions

In the RADseq derived trees *Deschampsia cespitosa* accessions of South Korea and Australia are found in a clade sister to the samples of New Zealand whereas the New Zealand *Deschampsia* endemics are sister to this entire *D. cespitosa* clade. This pattern is confirmed by the high co-ancestry within *D. cespitosa* (Figure 2A) and the clustering patterns (Appendix A and Figure 2B). After adding additional samples/taxa in the plastid tree, the *D. cespitosa* samples of New Zealand, Australia and Argentina appear in a clade with northern hemisphere samples of Europe, Canada, Russia, and Asia including *D. koelerioides* of China. This clade is sister to a southern hemisphere clade showing that the endemics share a common ancestor with *D. antarctica* of Antarctica and Argentina. Our data, therefore, point to a later arrival of the present *D. cespitosa*, clearly after the divergence of the endemics from an unknown ancestor. Considering the plastid data, one could assume vicariance of an ancestral southern lineage diverging into the New Zealand endemics and *D. antarctica*. However, vicariance can be ruled out given the geological history and the timing of the split from a common ancestor with *D. cespitosa* as estimated by the RAD data.

As in other regions the Pleistocene climatic oscillations have shaped the diversity and distribution of organisms in the Australis region. For Australia, Byrne et al. (2008) [87] argue that the phylogeographic patterns in southern Australia point to persistence and resilience rather than large-scale migration. In New Zealand, on the other hand, a variety of complicated patterns were found due to the diverse geological history of this region that behaves like a small continent [20].

The Asian *Deschampsia cespitosa* subsp. *orientalis*, subsp. *pamirica*, and *D. koelerioides* share a common ancestor, which may point to a scenario with *D. koelerioides* as a high alpine Central Asian endemic derivate of polymorphic *D. cespitosa*. Several taxa of Europe and Asia, including *D. koelerioides*, have sometimes been classified as a subspecies of *D. cespitosa* [88]. The taxonomical status of several endemic taxa in Europe and Asia is often debated [39,40,89].

*Deschampsia cespitosa* has always been considered native to New Zealand [45,50,85,90] although Allan (1936) [50] suspected that there could be a naturalized form present. There is no evidence for the introduction of *D. cespitosa* by the early British settlers in the 19th century, who brought with them agricultural and crop seeds, which were often contaminated by seeds of other species [91]. Indeed, plants native to or naturalized on the British Islands dominate the pool of alien plants in New Zealand. New Zealand *D. cespitosa* specimens were distinguished at varietal rank [49] based on awn insertion, but Edgar and Connor (2010) [45] rejected this and noted that plants did not differ from those of the Northern Hemisphere in their spikelet morphology. It is unclear as to the reasons of its decline in New Zealand, although intensive agriculture and wetland drainage are likely to be factors. Allan (1936) [50] stated, “is well-liked by stock in the young state” and de Lange (2020) [92] observed that it differs from those in other regions for being highly palatable, although Mark and Dickinson (2001) [38] observed no grazing effects at the ‘Pyramid’ lake population.

## 5. Conclusions

In this study we have combined a genomic approach with classical morphology including information on the life cycle to gain a better understanding of the evolutionary history of the genus *Deschampsia* in New Zealand. The high-resolution genomic data also allowed the correct identification of a hybrid population. Including samples from distant regions provided insight into evolutionary history on a global scale. We found evidence for at least two migration events of the genus *Deschampsia* to New Zealand. The native New Zealand populations of cosmopolitan *D. cespitosa* are members of a northern hemisphere clade, whereas the three endemic congeners belong to a southern hemisphere clade. They have diverged from a common ancestor with *D. cespitosa* in the late Miocene or Pliocene. Shifts in geological and climatical conditions with the onset of and during the Pleistocene are supposed to have triggered speciation within the endemic *Deschampsia* lineage. Although highly separated from *D. cespitosa* based on genomic and morphological traits, the endemics are involved in hybridization with the cosmopolitan congener.

## Figures and Tables

**Figure 1 biology-10-01001-f001:**
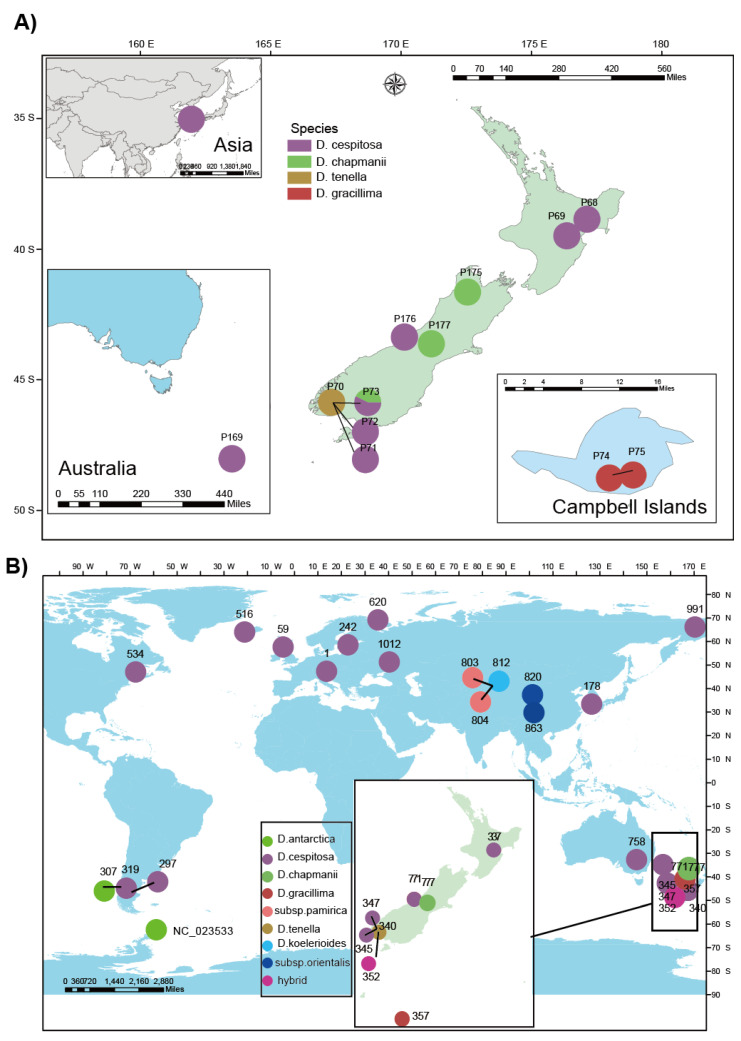
Sampling map for *Deschampsia* localities investigated in this study. (**A**) Sampling for RAD analysis; (**B**) Sampling for plastid genome analysis. Different species are highlighted with different colors. In map (**A**), populations of Korea, Australia and Campbell Islands are shown in inserted small maps. The coordinates of each population are given in Appendix A. The map data were downloaded from the open-access website: diva-gis.org (accessed on 12 October 2020).

**Figure 2 biology-10-01001-f002:**
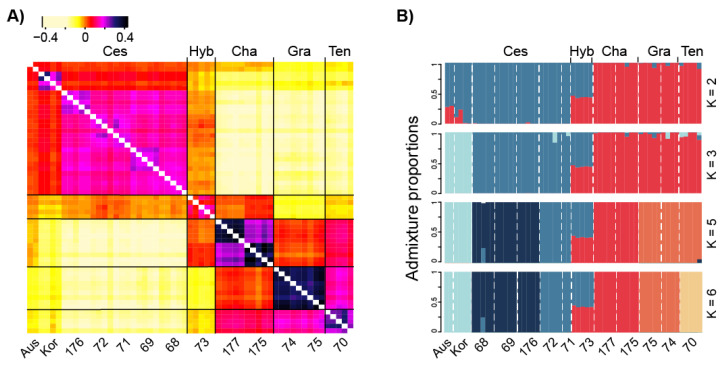
Clustering patterns of 57 accessions sampled across 13 sampling localities, based on 622,478 polymorphic sites filtered with ANGSD. (**A**) Heatmap obtained with covariance matrix results from PCAngsd. Population information is at the bottom and species names are on the top. Ces = *D. cespitosa*, Ten = *D. tenella*, Gra = *D. gracillima*, Cha = *D. chapmanii*. Hyb = Hybrid population, Aus = Australia population, Kor = Korea population. The diagonal estimation is excluded. (**B**) Genetic structure and admixture within *Deschampsia* in New Zealand and adjacent areas. Admixture diagrams are shown for K = 2, 3, 5, and 6. Ancestry proportions inferred with NGSadmix are shown as vertical bars where each vertical bar represents an individual. Species information is at the top of the figure. Population information is at the bottom. Species abbreviations are the same as in Heatmap.

**Figure 3 biology-10-01001-f003:**
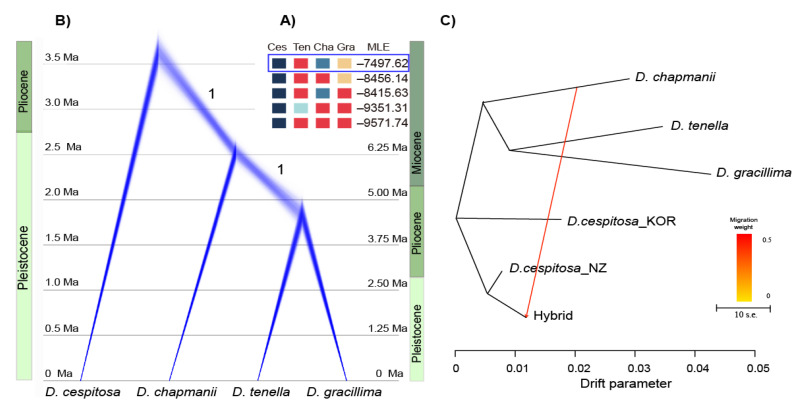
Results of coalescent-based phylogenetic inference. (**A**) Species delimitation models tests for *D. cespitosa* and three endemic species. Names of taxa follow Figure 2. MLE = marginal likelihood estimate. (**B**) The tree obtained for four *Deschampsia* species is shown as a cloudogram using Densitree. Posterior probabilities are presented for the relevant clades. The age was estimated by using average generation time estimates of *Deschampsia* (lower estimates for minimum two years, and upper estimates for maximum five years). (**C**) Treemix results based on 1600 unlinked SNPs in *Deschampsia,* excluding Australian *D. cespitosa*.

**Figure 4 biology-10-01001-f004:**
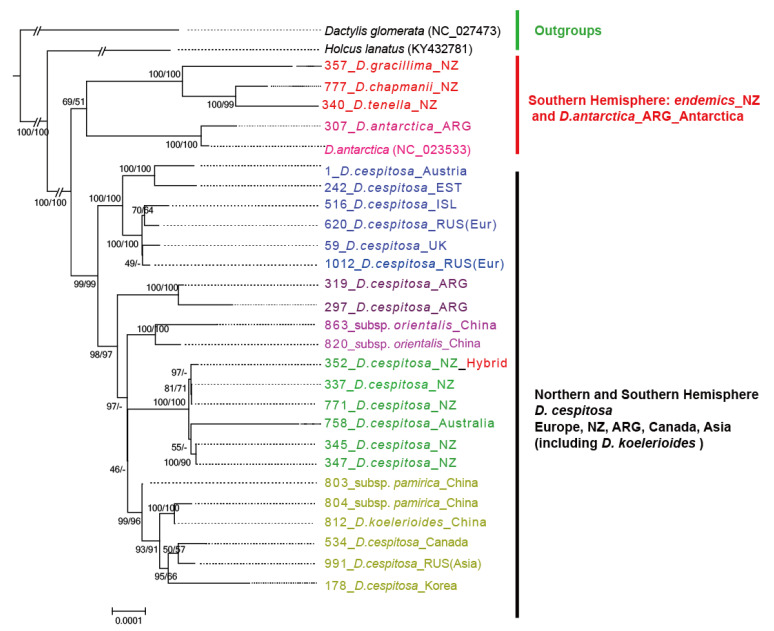
Maximum Likelihood phylogenetic tree based on whole plastid genome data. Both bootstrap of ML and Maximum Parsimony phylogenetic tree are shown on the branch of the ML tree. Both *Holcus lanatus* and *Dactylis glomerata* are used as outgroups. NZ = New Zealand, ARG = Argentina, EST = Estonia, ISL = Iceland, UK = United Kingdom, RUS = Russia, Eur = Europe.

**Figure 5 biology-10-01001-f005:**
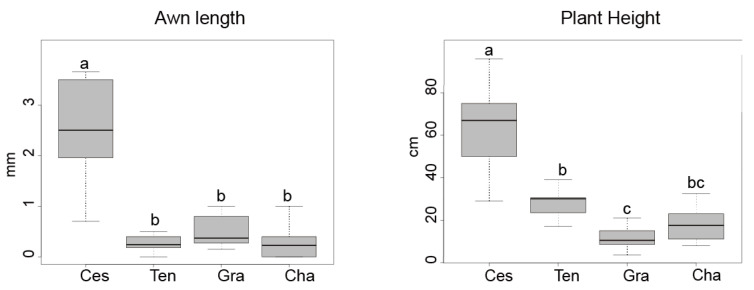
Boxplots showing variation in morphological traits in New Zealand *Deschampsia cespitosa* and three endemic taxa. Levels not connected by the same letter are significantly different according to the Mann–Whitney-U-tests (*p* < 0.05). Taxa codes: Ces = *D. cespitosa*, Ten = *D. tenella*, Gra = *D. gracillima*, Cha = *D. chapmanii*. *p* values of Mann–Whitney-U-tests for pairwise comparisons are shown in Table 1. Additional boxplots are given in Appendix A.

**Table 1 biology-10-01001-t001:** Bonferroni-corrected *p* values of Mann–Whitney-U-tests for pairwise comparisons between New Zealand *Deschampsia* taxa. Significant values <0.05 highlighted in bold.

Character	*Cesp/Chap*	*Cesp/Grac*	*Cesp/Tene*	*Chap/Grac*	*Chap/Tene*	*Grac/Tene*
Plant height	**0.002**	**0.001**	**0.023**	0.437	0.233	**0.005**
Basal leaf length	**0.001**	**0.001**	**0.004**	1.000	0.616	**0.039**
Penultimate leaf length	**0.003**	**0.002**	0.131	1.000	0. 112	0.065
Panicle length	**0.001**	**0.001**	**0.023**	0.862	0.604	**0.006**
Panicle width	0.141	0.056	1.000	1.000	0.126	**0.049**
Lower glume length	**0.001**	**0.001**	**0.003**	**0.002**	1.000	**0.004**
Upper glume length	**0.001**	**0.003**	**0.003**	**0.010**	1.000	**0.004**
Lemma length	**0.001**	**0.001**	**0.003**	**0.002**	**0.040**	0.986
Awn length	**0.007**	**0.003**	**0.005**	1.000	1.000	1.000

## Data Availability

RAD Sequence data and plastid genome alignment nex files for this study are available on request from the corresponding author.

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
