# Peer review of "The Evolutionary History of New Zealand Deschampsia Is Marked by Long-Distance Dispersal, Endemism, and Hybridization"

_biology, 2021, doi:10.3390/biology10101001_

Round 1

Reviewer 1 Report

I reviewed the manuscript “The evolutionary history of New Zealand Deschampsia is marked by long-distance dispersal, endemism, and hybridization”. The manuscript presents a huge amount of genomic data complemented with morphological information about Deschampsia species. I think that this manuscript represents a good contribution to the knowledge concerning evolutionary aspects of Deschampsia and it can be accepted to be published. I just have some minor suggestions to improve the final version.

Taxon Sampling:

Table S1: it was not clear for me why in cases for which was used 4 or 5 populations it is indicated only one geographic coordinate (lat/long). For example: I understood that for D. cespitosa from Korea four populations was used to obtain RadSeq data, however only one geographic coordinate was presented. Thus, I suggest to the authors a revision of this table to better clarify these informations.

In many places United Kingdom is cited as United Kingdome: please correct it.

RADseq evolutionary analyses:

Why did you fixed k = 9 to admixture estimation analysis? Could you better explain this choice?

Also, I would like to know why Hordeum vulgare was chosen as outgroup. I think that within Aveneae there are some other species with available genomic data.

Phylogenetic analyses for whole plastid genomes:

Holcus and Dactylis were used for rooting: why didn´t you used the same outgroup Hordeum for these analyses?

I would like to know also why the authors used Maximum Parsimony to phylogenetic reconstruction besides Maximum Likelihood.

Results:

I suggest to avoid some redundant terms as “monophyletic clade“: if it is a clade of course it is monophyletic.

In Fig. 3 I suggest to add the geological periods (Miocene, Pleistocene,...) to facilitate the discussion.

Figure 5: please correct Deschampsia in the caption.

Discussion:

In an overall way, I think that the discussion section was too summarized, I think that the data could be better exploited. The hypotheses of the study can be taken up in the light of the data obtained. I fell that the discussion concerning hybridization events is centered in the present but not in past events, this part could be complemented.  

Author Response

I reviewed the manuscript “The evolutionary history of New Zealand Deschampsia is marked by long-distance dispersal, endemism, and hybridization”. The manuscript presents a huge amount of genomic data complemented with morphological information about Deschampsia species. I think that this manuscript represents a good contribution to the knowledge concerning evolutionary aspects of Deschampsia and it can be accepted to be published. I just have some minor suggestions to improve the final version.

Taxon Sampling:

Table S1: it was not clear for me why in cases for which was used 4 or 5 populations it is indicated only one geographic coordinate (lat/long). For example: I understood that for D. cespitosa from Korea four populations was used to obtain RadSeq data, however only one geographic coordinate was presented. Thus, I suggest to the authors a revision of this table to better clarify these informations.

*** We have redone table S1, using “No. RAD” and “No. Plastid” to show individuals number included in RAD and chloroplast analysis. RADseq is based on the populations while plastid analysis is based on individuals. Besides, we use “Pop ID” and “Ind ID” to show the population and individual accession number in this study. In the RAD analysis, most populations have 5 individuals, several populations have 1-4 individuals. The Korea population has 4 individuals, which are in one population, with one geographic coordinate. Three populations have the same geographic coordinates (pop71, 72, 73), called ‘Pyramid’ lake site, which is the hybrid centre according to the morphological characters. So we collected more individuals here than in other regions. In plastid analysis, assuming this genome conserved, we chose only one individual from each RAD population. Besides, we included more samples from other regions. That’s why all individuals used for the plastid analysis have different latitude and Longitude.

 In many places United Kingdom is cited as United Kingdome: please correct it.

*** all corrected to United Kingdom

RADseq evolutionary analyses:

Why did you fixed k = 9 to admixture estimation analysis? Could you better explain this choice?

*** We ran the admixture estimation algorithm for several times to find the best K, which can reflect the true genetic group. If K is set to be extremely large, then the algorithm may easier get stuck and take longer execution time. If K is too small, the “true” structure cannot be detected. Our estimated number of clusters was based on the species (4), then we set higher than 4, firstly K=1-12, we found as K increased, the genetic groups can be separated based on the species and geographic regions, but when K was higher than 9, no more structure was found, then we chose K=1-9 for the admixture estimation and choose the best K according to Evanno’s ΔK statistic method. Fig. 2 shows Admixture diagrams for K = 2, and 3, 5, 6 which show also notably high ΔK peaks.

Also, I would like to know why Hordeum vulgare was chosen as outgroup. I think that within Aveneae there are some other species with available genomic data.

*** For the RADseq RaxML analysis, we needed to extract the outgroup SNP data which have the same positions with our own RAD data, since we didn’t include other outgroups in our RAD library during the lab work, we must extract the SNP information from Hordeum vulgare reference because we use this published genome during the mapping process. The reason why we used this reference is explained as follows: There is no Deschampsia genome available, we then downloaded the published genomes of Poaceae. Firstly, we used Denovo_map.pl in Stacks to assemble a reference ourselves and then compared the mapping quality using our own assembly genome and two published genomes in Poaceae (Hordeum vulgare and Triticum urartu), we found the Hordeum vulgare has higher mapping quality (33%) than the others (26.5% for own assembly genome and 20.5% for Triticum urartu), then we used Hordeum vulgare as the final reference. For the later analysis, the SNPs datasets from the mapped file both from STACKS and ANGSD program had a large number of informative sites and could give us enough information for different analyses.

Phylogenetic analyses for whole plastid genomes:

Holcus and Dactylis were used for rooting: why didn´t you used the same outgroup Hordeum for these analyses?

*** Because plastid sequences for those taxa closer to Deschampsia were available. For this analysis, we also tried different outgroups which are close to genus Deschampsia. We tried the following:  Anomochloa marantoidea (GQ329703); Holcus lanatus KY432781; Agrostis stolonifera NC_008591; Festuca ovina NC_019649; Dactylis glomerata NC_027473). The outgroups always had long branches and lead to very short branches in our ingroups. Then we chose two closer outgroups which could give us more information about the ingroups. 

I would like to know also why the authors used Maximum Parsimony to phylogenetic reconstruction besides Maximum Likelihood.

*** To get an idea on robustness of data. Maximum Likelihood tree is based on the maximum likelihood between genetic data while Maximum Parsimony tree is based on the minimal number of character state changes. The real gene sequences evolve heterogeneously and are not identically distributed, which may give us biased and inconsistent results. When these two techniques are used for tree construction, the reliability and accuracy are high. The topologies of MP and ML tree were congruent in our result, which means that the relationship in our chloroplast analysis is reliable and accurate.  

Results:

I suggest to avoid some redundant terms as “monophyletic clade“: if it is a clade of course it is monophyletic.

*** now corrected

In Fig. 3 I suggest to add the geological periods (Miocene, Pleistocene,...) to facilitate the discussion.

*** Done

Figure 5: please correct Deschampsia in the caption.

*** Done

Discussion:

In an overall way, I think that the discussion section was too summarized, I think that the data could be better exploited. The hypotheses of the study can be taken up in the light of the data obtained. I fell that the discussion concerning hybridization events is centered in the present but not in past events, this part could be complemented.

*** We have slightly modified the discussion.

Regarding the hybrid we have added in 4.2:

“From our data (nearly balanced proportions in Fig. 2B) we conclude that this is a rather recent hybridization because for an ancient introgression we would expect a shift towards the dominant taxon.”

Regarding the southern lineage we have added in 4.3:

"Considering the plastid data, one could assume vicariance of an ancestral southern lineage diverging into the New Zealand endemics and D. antarctica. However, vicariance can be ruled out given the geological history and the timing of the split from a common ancestor with D. cespitosa as estimated by the RAD data."

Reviewer 2 Report

Manuscript review of Xue et al., “The evolutionary history of New Zealand Deschampsia.....”, for Biology
(MDPI), 16 September 2021.

The authors use RAD sequence markers, whole chloroplast genome sequencing, and morphological
analysis to explore the origins and relationships of 4 Deschampsia species in New Zealand. The
manuscript is very well written with clear goals, methods, and interpretation of data. I have just a few
minor suggestions for revision:

Line 22: Insert ‘We hypothesize that’ the endemics diverged..........;

Line 62: Use ‘subsequent’ instead of ‘posterior’

Line 166: Provide a statement as to why the fourth endemic species (D. pusilla) was excluded from study

Line 525: New Zealand endemics ‘plus D. antarctica’ are sister......

Line 562: Here it is specified that the endemic taxa arrived by migration to New Zealand. Given their
relationship to D. antarctica and the timing of divergence events, is vicariance involving a widespread
ancestral Southern taxon possible? If not, perhaps specify in the discussion that vicariance is ruled out,
and why. This would tie in nicely with the discussion of dispersal and vicariance emphasized in the
introduction.

Author Response

Manuscript review of Xue et al., “The evolutionary history of New Zealand Deschampsia.....”, for Biology (MDPI), 16 September 2021.

The authors use RAD sequence markers, whole chloroplast genome sequencing, and morphological analysis to explore the origins and relationships of 4 Deschampsia species in New Zealand. The manuscript is very well written with clear goals, methods, and interpretation of data. I have just a few minor suggestions for revision:

Line 22: Insert ‘We hypothesize that’ the endemics diverged..........;

*** Done

Line 62: Use ‘subsequent’ instead of ‘posterior’

*** Done

Line 166: Provide a statement as to why the fourth endemic species (D. pusilla) was excluded from study

*** We have added: “Several attempts to collect D. pusilla failed.”

Line 525: New Zealand endemics ‘plus D. antarctica’ are sister......

*** No, we cannot add D. antarctica here because these first sentences are dealing only with RADseq data (no data of D. antarctica included in RADseq). See also Figs. 1A, 2, 3).

Line 562: Here it is specified that the endemic taxa arrived by migration to New Zealand. Given their relationship to D. antarctica and the timing of divergence events, is vicariance involving a widespread ancestral Southern taxon possible? If not, perhaps specify in the discussion that vicariance is ruled out, and why. This would tie in nicely with the discussion of dispersal and vicariance emphasized in the introduction.

*** We say that the endemics share a common ancestor with D. antarctica. And we have some idea on the divergence time (late Miocene/Pliocene) of the endemics (the southern lineage) from ancestral D. cespitosa, but we do not know when and how this southern lineage did arrive on New Zealand. However, vicariant speciation is unilkely given divergence timing and geological history. Therefore, we have added:"Considering the plastid data, one could assume vicariance of an ancestral southern lineage diverging into the New Zealand endemics and D. antarctica. However, vicariance can be ruled out given the geological history and the timing of the split from a common ancestor with D. cespitosa as estimated by the RAD data."